# Influence of COVID-19-Related Restrictions on the Prevalence of Overweight and Obese Czech Children

**DOI:** 10.3390/ijerph191911902

**Published:** 2022-09-21

**Authors:** Anna Vážná, Jana Vignerová, Marek Brabec, Jan Novák, Bohuslav Procházka, Antonín Gabera, Petr Sedlak

**Affiliations:** 1Department of Anthropology and Human Genetics, Faculty of Science, Charles University, Albertov 6, 128 00 Prague, Czech Republic; 2Institute of Endocrinology, Národní 8, 110 00 Prague, Czech Republic; 3Institute of Computer Science, Czech Academy of Sciences, Pod Vodárenskou Věží 271/2, 182 00 Prague, Czech Republic; 4National Institute of Public Health, Srobarova 48, 100 00 Prague, Czech Republic; 5MUDr Bohuslav Procházka s.r.o., Radnická 635, 284 01 Kutná Hora, Czech Republic; 6Zdravotní Středisko Krásné Březno, U Pivovarské Zahrady 5, 400 07 Ústí nad Labem, Czech Republic

**Keywords:** COVID-19, children, obesity, severe obesity, COVID-19-related-restrictions effect, GAM, semiparametric statistical modeling

## Abstract

Apart from influencing the health of the worldwide population, the COVID-19 pandemic changed the day-to-day life of all, including children. A sedentary lifestyle along with the transformation of eating and sleep habits took place in the child population. These changes created a highly obesogenic environment. Our aim was to evaluate the current weight in the child population and identify the real effects of the pandemic. Height and weight data were collected by pediatricians from the pre-COVID-19 and post-COVID-19 periods from 3517 children (1759 boys and 1758 girls) aged 4.71 to 17.33 years. We found a significant rise in the z-score BMI between pediatric visits in the years 2019 and 2021 in both sexes aged 7, 9, 11, and 13 years. Especially alarming were the percentages of (severely) obese boys at the ages of 9 and 11 years, which exceed even the percentages of overweight boys. With the use of statistical modeling, we registered the most dramatic increment at around 12 years of age in both sexes. Based on our research in the Czech Republic, we can confirm the predictions that were given at the beginning of the pandemic that COVID-19-related restrictions worsened the already present problem of obesity and excess weight in children.

## 1. Introduction 

The spread of SARS-CoV-2 infection, first reported in Wuhan, China, in December 2019, has challenged the whole world [1]. The illness’s manifestations vary from mild flu-like symptoms to life-threatening states [2]. Due to the spread and critical epidemiologic situation, COVID-19-related restrictions were introduced all around the world. In addition to the many changes in collective activities and events, the home-office was established for many people; furthermore, schools were closed to ensure the limitation of the spread of COVID among children in many countries around the world. 

The numerous COVID-19-related restrictions that shaped the whole of society also had a very specific impact on the child population. Along with the regular use of electronic devices and social platforms, e-learning and connecting via social media with friends and family took place even more; thus, the screen time of children has risen dramatically. A Spanish study showed that screen time has risen on average by +114 min/day. On the other hand, physical activity declined by 91 min/day [3]. Children did not have to commute to school and organized sports were suspended. Combined with the transition to a sedentary lifestyle, children furthermore lacked the structured alimentation consisting of nutritionally balanced meals and consumed more snacks during the day. The impact of COVID-19-related restrictions resulted not only in online education but also in the limitation of physical activity, thus changing the lifestyle of the whole population. Nutrition and lifestyle have been altered and challenged. Even before the current situation, the numbers of overweight and obese children were substantial. In 1975, was 6 million boys and 5 million girls with obesity between the ages of 5 and 19 years. In 2016, the numbers increased to 75 million obese boys and 50 million girls [4].

Even though the situations in different countries and parts of the world varied, we can anticipate significant changes in alimentation due to different causes. Due to complicated situations, families could have lost or had reduced income, which could have led to the consumption of less nutritious food for its lower cost [5]. Moreover, due to the difficult situation, people tended to buy less fresh food and buy more ultra-processed food that has long durability and is convenient [6]. Stress that was created for example by a bad financial situation in the family and a restricted social life could have led to craving comfort food that usually contains excess sugar and/or fat [7]. Along with changes in alimentation, there are the changes in physical activity. As a result of the restrictions to attending school and sport facilities, children tended to engage in sedentary behavior [8]. It is a well-known fact that children without a regular schedule, comprising of insufficient physical activity, irregularity in the sleep schedule, and sedentary behavior, all with higher food intake, gain weight. These factors occur predominantly during the summer, when it was proven that children gain the most weight during the year [9,10,11,12]. Some studies even calculated that the prevalence of overweight and obesity regularly rises during the summer school holidays by 2–3% [13]. 

With this evidence, we can presume that the number of children being overweight or obese has risen during the societal changes resulting from the COVID-19-related restrictions. There has even been evidence that the situation of already overweight or obese children has worsened over the pandemic period [11]. The trend of increasing the frequency of obesity in children was evident even before the COVID-19 situation. The rising prevalence of obesity and state of being overweight has been more pronounced in pre-puberty and puberty, and more in boys [14,15]. It is accurate to state that the world is facing not only one pandemic but actually two: COVID-19 and obesity. Due to the previously mentioned factors, the two epidemics interact, and the obesity situation can be expected to become progressively worse. It is important to monitor the obesity problem in order to devise effective counteractive and mitigation measures in the post-epidemic period. 

The main aim of our study was to contrast the profile of age versus the prevalence of overweight and obese children before and shortly after the pandemic. We also aspired to evaluate the long-term trend of obesity and prevalence of overweight children. Based on semi-parametric modeling, we intended to investigate the effect of the COVID-19 lock-down on the development of the prevalence of obese and overweight Czech children and adolescents, both in absolute terms and in the redistribution (or deformation) of obesity risk across age in childhood and adolescence.

## 2. Materials and Methods

### 2.1. Background

The proclamation of COVID-19 as a pandemic on 12 March 2020 by the WHO (World Health Organization) was one day after the introduction of COVID-19-related-restrictions in the Czech Republic, which included the closure of schools on 11 March; the state of emergency that was announced in the Czech Republic followed on 12 March. Although restrictions for elementary schools were reduced at the end of April 2020, Czech children were home-schooled/on-line taught for approximately two and half months that spring (11 March–25 May 2020). The lifting of heavy restrictions did not mean that school and other activities were back to normal. The bans and restrictions in schools and also in sport and other activities were not differentiated as to whether they were collective or individual activities. After spring of 2020, the two-month long school summer holidays took place as usual. In September, even though the children started their school year normally, another heavy set of restrictions were imposed due to the worsening epidemiological situation, resulting in one and a half more month of on-line schooling during autumn 2020 (14 October–30 November). The situation during spring 2021 resulted in another two-month closure (27 February–26 April) of educational institutions in the Czech Republic.

### 2.2. Data Collection

The design of this study was prospective with the incorporation of retrospective data collection. There were 63 pediatricians participating in the study, providing data on 3517 children (1759 boys and 1758 girls). Their ages at the most recent data entry were between 4.71 and 17.33 years. The collection of data was conducted by pediatricians across the country with representation from 13 out of the 14 regions of the Czech Republic. These children underwent standard examinations by Czech pediatricians within compulsory preventive examinations that were carried out on children at the ages of 1.5, 3, 5, 7, 9, 11, 13, 15, and 17 years. This biennial examination consists of a check-up of the health condition of a child, including measurements of height, weight, blood pressure, and a clinical examination. These standard examinations are conducted around certain birthdays of a child with a dispersion ± 4 months (e.g., an examination of a 5-year-old is done between the ages of 4 years 8 months and 5 years 4 months). 

This project took place during the summer of 2021, with the most recent data entry set being between April and the end of June 2021. The data consist of the latest examination and up to 3 entries of examinations that were done in the past, conventionally biennially (i.e., the database is compiled of data from the years 2021, 2019, 2017, 2015), thus from the period before the COVID-19 pandemic. The data from past examinations were used for the completion of the growth and development of children, to use to create a model that objectively evaluates the effect of COVID-19-related restrictions and lockdowns. The data were entered into a unified database and thoroughly checked by a single person (expert anthropologist). Possible errors that were based on the inspection of minimal and maximal values, as well as the gradual development of children, were examined and verified on a case-by-case basis. 

### 2.3. Measurement

The data that were collected by pediatricians consist of date of birth, measurements of height (in centimeters, to one decimal), weight (in kilograms, to one decimal), and body mass index (BMI), as a standard approach during pediatric examinations. The measurements of weight and height were obtained using standard equipment of medical facilities, that are calibrated and certified, with standardized approach and precision. BMI was computed as BMI = kg/m^2^. Due to the specific BMI values for children, a z-score (SD) of BMI was applied, in accordance with WHO references for children [16]. BMI categories were defined according to WHO from −2 to 1 SD normosthenic, >1 SD overweight, >2 SD obesity, and >3 SD severe obesity [16]. 

### 2.4. Statistical Analyses

The data were processed in Microsoft Excel and properly anthropologically checked. The z-scores of anthropometric data were evaluated in Anthro and Anthro Plus, software that was produced by WHO (https://www.who.int/tools/child-growth-standards/software, accessed on 20 May 2022). Statistical analyses were carried out in EpiData Analysis V. 2.2.3.187 (www.epidata.dk, accessed on 20 May 2022). 

Differences in the BMI z-scores for each year (re-test), age group, and sex were calculated using a Durbin–-Conover post hoc test in Jamovi 2.3.15 software (https://www.jamovi.org/, accessed on 20 May 2022).

In order to investigate the effect of COVID-19-related restrictions and societal changes in detail, we formulated a flexible semiparametric GAM (generalized additive model) [17,18]. Our model aims to primarily detect the COVID-19-related-restriction effects on the mean BMI for both sexes separately. Our preliminary hypothesis is that these effects, if present, can possibly be quite different for different ages. We hypothesize that they will be smaller for very young children as well as for the oldest children/youth just before adulthood. To this end, we have to allow for an age-differential COVID-19-related restrictions effect. Moreover, to evaluate the differential, we must have a solid baseline (pre-COVID-19 years). In order to exclude the need to rely on external references (such as the now old national standards) and hence the need to assume perfect representativity of our sample with respect to the population as a whole (which we probably do not have), we derive our study internal baseline as a part of the statistical model. Since we do not know the exact shape of either the age-different COVID-19-year effect, or the age baseline trend, we have to formulate both components non-parametrically—as unknown smooth curves. We implement them as a penalized spline [19]. Furthermore, we have to account for autocorrelation that is induced by the repeated measurement of the same individual. To this end, we include the random [20] individual-specific individual effect. Effectively, our model is then a GAMM (generalized additive mixed model), but computationally we treat it as a special case of GAM formulation with REML (restricted maximum likelihood) estimation of unknown penalty constants (including random effect variance). In detail, our model is:$$Y_{ita}=\mu +b_{i}+s_{age}\left(a\right)+s_{dif}\left(a\right)\times I\left(t=2021\right)+\varepsilon_{ita}$$
where:$Y_{ita}$ is the BMI of child *i* that is measured in year *t* at age *a*
μ is an overall intercept (unknown parameter to be estimated from data)$b_{i}$ is the random effect of the *i*-th child ($b_{i}\sim N\left(0,\sigma_{b}^{2}\right)$)$\varepsilon_{ita}$ is the random error of child *i* that is measured in year *t* at age *a* ($\varepsilon_{i}\sim N\left(0,\sigma^{2}\right)$)$s_{age}$ is the baseline (pre-COVID-19 year) mean BMI age profile (unknown smooth function to be estimated from the data)$s_{dif}$ is the age-differential increment in BMI in COVID-19-year (unknown smooth function to be estimated from the data)I(.) is an indicator function (assuming the value of 1 if its argument is true and the value of 0 otherwise)


The model is fitted (i.e., identified by the estimation of all unknown coefficients and smooth functions) via penalized likelihood [18] with the unknown penalty coefficients estimated via REML [19]. The computations are done in the mgcv library [18] in an R statistical environment (R core team 2022). 

### 2.5. Ethical Considerations

The present research was approved by the Institute of Endocrinology, Národní 8, 110 00 Prague 1, Czech Republic. The participation of the children was subject to informed permission of the parents or legal representatives of the children. All procedures contributing to this work were complied with the ethical standards of the relevant national and institutional committee on human experimentation and with the Helsinki Declaration of 1975, as revised in 2008. 

## 3. Results

Figure 1 and Figure 2 present the prevalence of overweight, obese, and severely obese children in each age group. One of the most concerning results of our study is the ratio between overweight and (severely) obese children. At the ages of 7 and 13 years, the ratio between overweight and (severely) obese boys is about equal. At the age of 7 years, the ratio is 17.1% and 16.0%, respectively. At the age of 13 years, the ratio is 23.1% and 22.7%. However, the ratio of the pre-pubertal boys reverses and at the age of 9 years the ratio is 17.6% and 21.5%, respectively; at the age of 11 years the ratio is 21.8% and 27.8%. The same problem occurred in 2019 in boys of 11 years where the ratio between overweight and (severely) obese was 15.7% and 19.7% (see Appendix A). The present prevalence of overweight and obese children in all age groups is considerable. The highest numbers are seen at the age of 11 and 13 in both boys’ and girls’ groups. The escalating tendency of prevalence is also seen in younger age groups. The numbers of children in the category of severe obesity especially present the most serious problem, most notably in the boys at the ages of 7 and 9, where these categories account for more than half of the total number of obese children. From the point of view of prevalence as well as severity of obesity, the situation is more concerning in all age groups of boys. 

The data in Table 1 monitor the BMI z-score progress from different pediatric visits of children divided into categories according to their age in 2021. The structure of visits is described in the Materials and Methods section. Table 1 presents the differences between each visit to show if there were significant changes. Using the Durbin–Conover post hoc test for comparing the BMI z-scores, we observed a significant and consistent increase of the BMI z-scores between the visits in 2019 and 2021, especially among boys and girls between 7 and 13. A similar table (Table A1) with values of BMI is shown in the Appendix A. 

The semiparametric GAM model that we use enables us to derive a study-specific average mean BMI age profile for the pre-COVID-19 years. Therefore, we do not have to assume perfect population distribution coverage for our study sample (which would be both necessary and hard to achieve if we had to rely on population standards). Figure 3 and Figure 4 show these estimated pre-COVID-19 mean BMI age trends together with their 95% confidence limits for boys and girls. The same model is able to estimate differential (age-specific) increments of BMI in the time of COVID-19-related restrictions, hence showing the net COVID-19 effects (separated from the smooth secular obesity trend in the pre-COVID-19 years) and demonstrating that it is markedly different for the various ages. Both the differential COVID-19-year effect and age trends are highly significant for both sexes (*p*-value < 0.001). As shown in Figure 5 and Figure 6 for boys and girls, the increment is most dramatic at around 12 years of age. The peak location is similar for both ages, as is the overall size of the effect. For boys, it is significant for all ages except for small children (up to 4 years of age). For girls, it is also not significant for small children up to 4, as well as for the oldest ages. 

## 4. Discussion

The pandemic situation brought up many problems, especially at the beginning when the main focus was oriented to managing new challenges in public health. However, many researchers expressed concern about the secondary effects of lockdown and restrictions that could result in various issues [21,22,23,24]. Many have stated that the COVID-19 pandemic collided with another pandemic that was already here for a while: the pandemic of obesity and overweight [25,26].

The growth of the prevalence of obesity has caused concern for many. Due to the COVID-19 situation and required distancing, research had to be limited and mainly organized with a small number of probands or conducted via questionnaires and self-reporting. The prevalence of these research results was concerning. In Greece, elaborate questionnaires for parents brought considerable data from almost 400 children/adolescents about sleep, screen time, and physical activity, along with body weight. The increase of weight was described ins 35% of children/adolescents and physical activity dropped by 66.9% [27]. Similar to our concept of sharing data from pediatricians, a Massachusetts study with more than 46,000 children showed that the prevalence of obesity (≥95th percentile of BMI) rose from 15.1% in 2018 and 15.7% in 2019 to 17.3% in 2020. A greater increase was seen in boys aged 6 to 11 years [28]. Parallel research that was done in cooperation with pediatricians was conducted in the Philadelphia region with 300,000 patients of age 2–17 years. The results showed a rise of prevalence in obesity across all ages [29]. Moreover, the problem of already obese children has worsened [11]. 

Our findings correspond with the findings of the Massachusetts study, where boys between 6 and 11 years had a major rise in obesity. Czech boys in our study in the ages of 9 and 11 years show concerning numbers in obesity and severe obesity, whose combined numbers exceed the percentage of overweight boys. This finding is potentially alarming in its health impact on their current state of health, as well as the future health of these boys. When comparing the visits from 2021 and 2019, the data of the z-score of BMI show that in the final age categories of 7, 9, 11, and 13 years in both sexes weight gain had significantly risen. As the last nationwide anthropological research on children and youth in the Czech Republic took place in 2001, it is not suitable to use for the comparison of pre- and post-COVID-19 states of obesity in children. Nevertheless, in the Czech Republic the situation was evaluated on 7-year-old children between 2015 and 2017 within the WHO project COSI showing 22.9% overweight or obese boys and 19.1% girls [30]. Appendix A
Table A2 compares our findings from 2021 and 2015 with those of COSI 2015-17. Our results from visits in 2015 correspond with the COSI results and affirm our results. Our data modeling based on our own dataset separates the trend of the increasing prevalence of weight gain in children that was present even before the pandemic and the leap that occurred in the reaction to the obesogenic environment that was created by the COVID-19 pandemic in the child population of the Czech Republic. 

Within the focus on the increase of obesity, the authors are also looking into the mechanisms that are accountable for the existing obesogenic environment. With the lockdown and social distancing restrictions, fresh food may have been substituted with more processed foods containing more sugar and fat. This phenomenon was due to a variety of reasons such as food insecurity and convenience [31,32,33]. More children also tended to snack more and whole nutritional alimentation was shifted towards frequent eating [31,34]. In modern society, children are encouraged to join organized sports and activities. As structured sports were banned, the children who mainly take part in such activities were severely impacted and lacked overall physical activity more than children from rural areas who are not as dependent on structured and organized activities as children from urban areas [35]. That corresponds with the idea that active transport (i.e., walking and biking) and independence of structured sports is beneficial for maintaining physical activity. When the main part of physical activity is attributed to organized sport, any bans or restrictions will severely influence the physical movement of many. It was generally documented in 10 European countries that the physical activity recommendations were met by less than 20% of children and adolescents during the pandemic [36]. Moreover, much more screen time, which goes hand-in-hand with sedentary behavior, took place. This was necessitated by the online schooling that took place. Nevertheless, the amount of time that children spent in their free time using screens (for amusement, relaxation, and also as a distraction so parents could work from home) was closely observed. In Germany, screen time rose in children and adolescents by 37 to 79 minutes daily [37]. In Canada, it was confirmed that school-aged children also increased their recreational screen time [38]. 

The rise of weight gain in prepubertal children brings consequences that are connected to the alteration of the puberty period. The link between fat tissue and menarche is well known. Excess fat tissue and obesity have been linked with the acceleration of growth and timing of puberty [39,40]. The endocrine effect of leptin, which is produced mainly from fat tissue and whose high levels suppress food intake, is deteriorated in obese individuals, who are leptin-resistant. Another effect of leptin is the above-mentioned acceleration of puberty and as a secondary result the acceleration of growth during that pubertal growth spurt. [41,42]. Moreover, apart from the alteration of growth and puberty, hypogonadism can be gradually developed in obese boys. The suppression of the hypothalamic-pituitary-testicular axis is affected by leptin. Furthermore, the pro-inflammatory cytokines from fat-tissue, which affect this axis and secondarily suppresses the levels of free testosterone and/or estradiol, results in clinical androgen deficiency. An additional effect of fat-tissue in males is adipose estrogenization, resulting in habitual feminization such as a broader pelvis [43]. 

In the context of obesity and COVID-19-related restrictions during the pandemic, the prognosis for the future is disturbing. In the globalized world, the quick spread of infectious diseases at high rates could possibly result in an overload of medical facilities and higher rates of death. Individuals who are overweight or obese are at higher risk of hospitalization, due to altered immunity as a consequence of excess fat tissue [44]. It has also been shown that the effectivity of vaccination is compromised in people with obesity or excessive weight [45]. Obesity that starts in childhood/adolescence has a great risk of persisting into adulthood as well [46,47]. The restrictions that followed the pandemic situation encouraged obesogenic behavior, creating a further threat to already overweight and obese children, as well as to children who had had normal weight until then. Children and adolescents as well as adults got into an obesogenic environment that may possibly have resulted in the increase of overweight and obesity, leading to additional health issues. In the context of the current situation, the term “covibesity”, the merging of two pandemics that collided and combined, is accurate.

## 5. Conclusions

The trend of obesity and overweight in children is a grave problem that escalated due to the obesogenic environment that the COVID-19 pandemic brought. Our concern is about the high percentages in prepubertal boys, where obesity and severe obesity even exceeded the percentages of overweight boys. Our exploration shows that the pandemic aggravated the obesity problem that already existed in our society even before the pandemic. With the use of statistical modeling, we recognized the biggest effect of pandemic restrictions on weight gain in both boys and girls around the age of 12. What is especially concerning is the prevalence of (severely) obese versus overweight boys, which was unprecedented. We found a significant rise in the z-score BMI between visits in 2019 and 2021 in both sexes of the final age of 7, 9, 11, and 13 years. There were particularly alarming percentages of (severely) obese boys at the ages of 9 and 11 years, exceeding the percentages of merely overweight boys. With the use of statistical modeling, we registered the most dramatic increments at around 12 years of age in both sexes.

Among other future serious health issues of each overweight/obese individual is the very likely acceleration and alteration of puberty due to the primary and secondary effects of fat-tissue. For boys, another threat is the slowly developing hypogonadism that originates in obesity. Based on our research, we can confirm the predictions that were made at the beginning of 2020, that the pandemic will worsen the already present problem of obese and overweight children, which will bring many challenges in the future.

## Figures and Tables

**Figure 1 ijerph-19-11902-f001:**
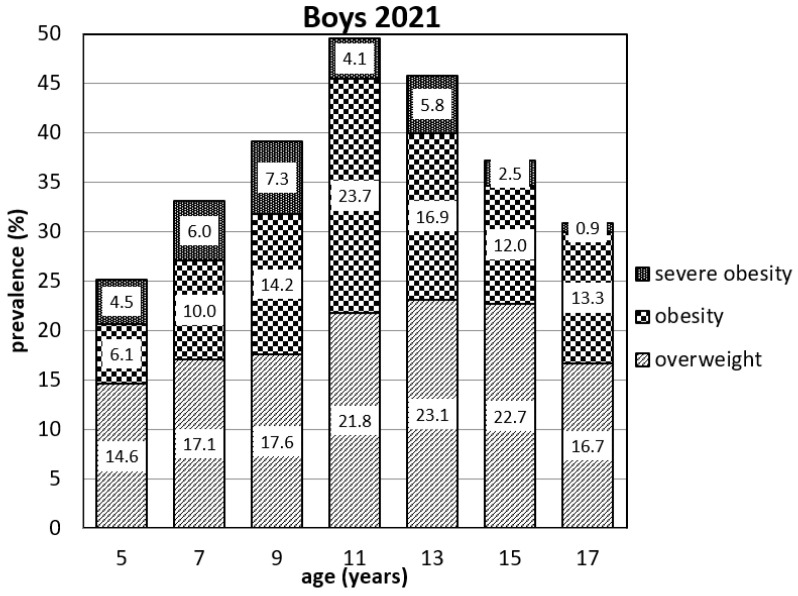
Prevalence of overweight, obese, and severely obese boys in 2021.

**Figure 2 ijerph-19-11902-f002:**
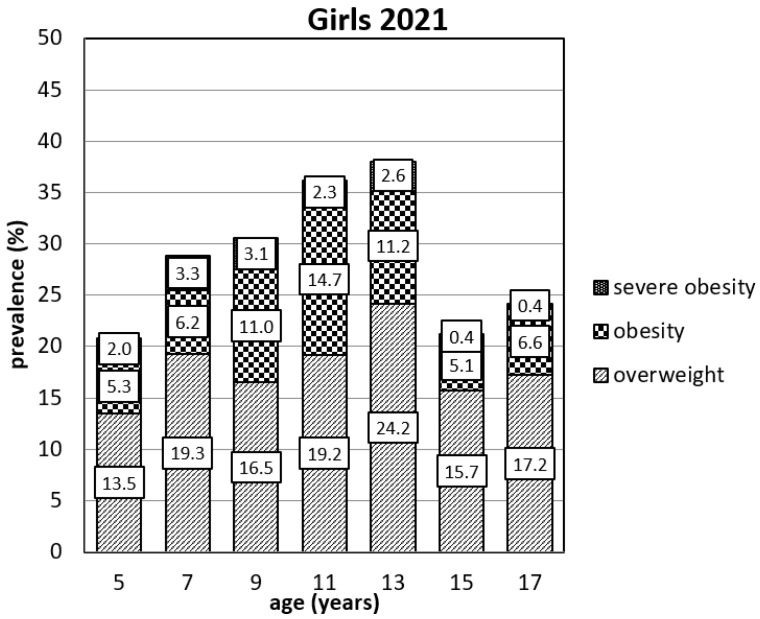
Prevalence of overweight, obese, and severely obese girls in 2021.

**Figure 3 ijerph-19-11902-f003:**
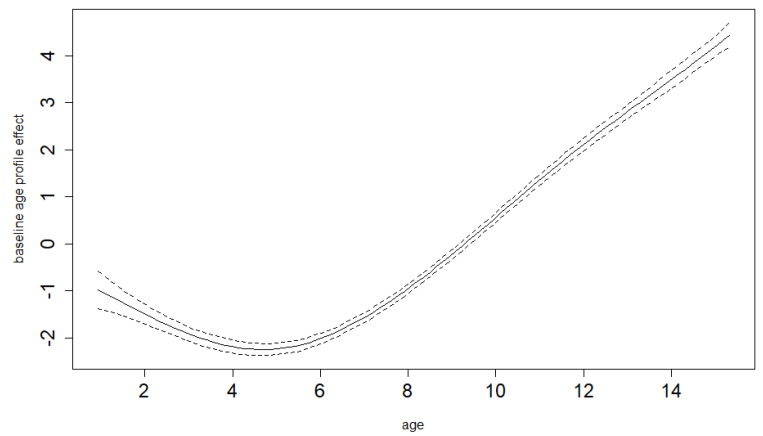
Baseline BMI age profile for boys, estimated smooth trend (solid line) and pointwise 95% confidence interval limits (dotted lines). Age in years.

**Figure 4 ijerph-19-11902-f004:**
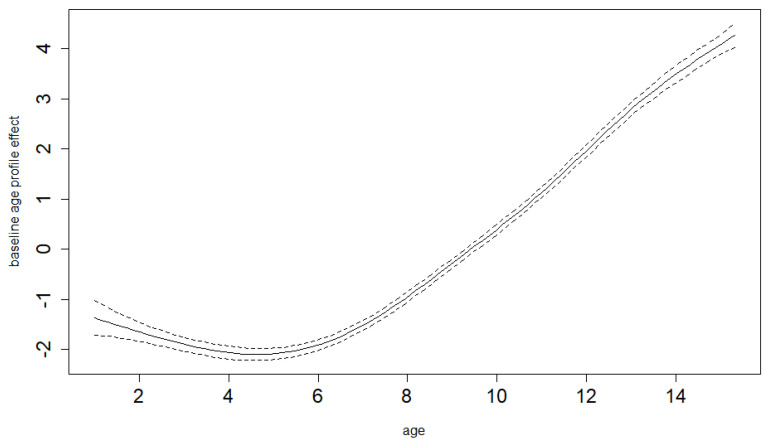
Baseline BMI age profile for girls, estimated smooth trend (solid line) and pointwise 95% confidence interval limits (dotted lines). Age in years.

**Figure 5 ijerph-19-11902-f005:**
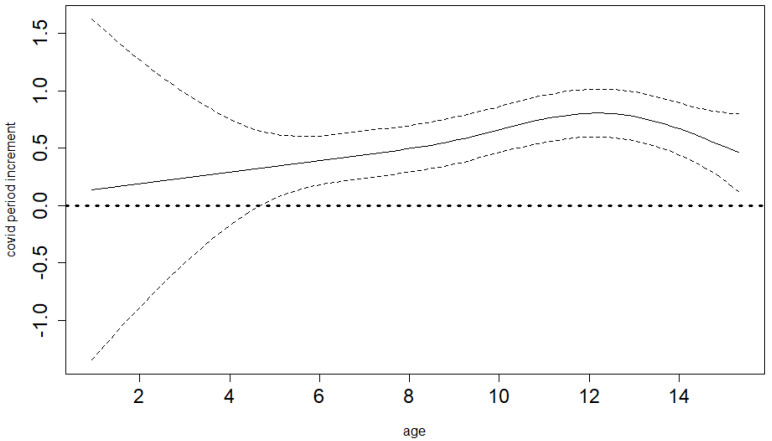
Differential COVID-19 year effect for different ages for boys, estimated smooth trend (solid line) and pointwise 95% confidence interval limits (dotted lines). Age in years.

**Figure 6 ijerph-19-11902-f006:**
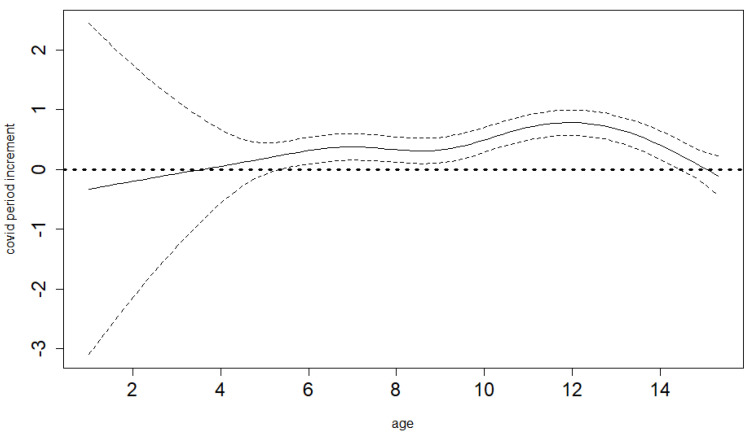
Differential COVID-19-year effect for different ages in girls, estimated smooth trend (solid line) and pointwise 95% confidence interval limits (dotted lines). Age in years.

**Table 1 ijerph-19-11902-t001:** Mean values, number of probands, and the standard error of the mean (SEM) for each age group and sex. Differences between re-tests were calculated using Durbin-Conover post hoc test (D-C).

Final Age (y)		2021 Visit			2021–2019 Diff		2019 Visit			2019–2017 Diff		2017 Visit			2017–2015 Diff		2015 Visit		
	Sex	n	Mean	SEM	D-C	*p*-Value	n	Mean	SEM	D-C	*p*-Value	n	Mean	SEM	D-C	*p*-Value	n	Mean	SEM
5	boys	246	0.42	0.09	1.94	0.054	244	0.21	0.07	1.72	**0.086**	98	0.03	0.10			0		
	girls	245	0.16	0.08	0.44	0.656	242	0.13	0.07	1.30	0.193	107	0.33	0.12			0		
7	boys	251	0.56	0.10	3.67	**0.0003**	250	0.13	0.08	2.25	**0.024**	240	0.19	0.09	2.17	**0.031**	89	0.34	0.12
	girls	243	0.40	0.08	2.61	**0.009**	240	0.02	0.07	1.89	**0.060**	232	0.06	0.07	0.89	0.373	67	0.24	0.13
9	boys	261	0.70	0.09	6.96	**<0.00001**	261	0.23	0.09	0.50	0.615	253	0.10	0.07	1.31	0.189	224	0.16	0.07
	girls	254	0.4	0.09	3.73	**0.0002**	251	0.19	0.08	0.05	0.954	248	0.15	0.08	1.33	0.182	215	0.15	0.08
11	boys	266	1.01	0.09	5.76	**<0.00001**	265	0.68	0.08	2.97	**0.003**	260	0.43	0.08	0.21	0.831	232	0.23	0.07
	girls	266	0.55	0.08	3.55	**0.0004**	261	0.36	0.08	2.07	**0.038**	254	0.22	0.07	2.03	**0.041**	235	0.05	0.07
13	boys	260	0.72	0.10	3.11	**0.002**	254	0.56	0.09	1.07	0.280	251	0.45	0.09	3.3	**0.001**	232	0.16	0.10
	girls	269	0.56	0.07	3.02	**0.002**	266	0.40	0.07	0.85	0.390	265	0.34	0.07	2.72	0.006	231	0.21	0.08
15	boys	242	0.53	0.08	1.31	0.190	237	0.43	0.09	1.10	0.270	233	0.45	0.08	2.09	0.036	222	0.33	0.09
	girls	254	0.09	0.07	0.52	0.590	252	0.11	0.07	0.68	0.490	245	0.05	0.07	0.05	0.950	232	0.01	0.07
17	boys	233	0.43	0.08	0.98	0.320	231	0.49	0.08	0.64	0.510	227	0.52	0.09	2.35	**0.010**	214	0.64	0.09
	girls	227	0.18	0.07	0.94	0.340	222	0.18	0.07	1.53	0.120	218	0.22	0.08	2.32	**0.020**	208	0.13	0.09

## Data Availability

The data presented in this study are available on request from the corresponding author.

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
