# Peer review of "Influence of COVID-19-Related Restrictions on the Prevalence of Overweight and Obese Czech Children"

_ijerph, 2022, doi:10.3390/ijerph191911902_

Round 1
Reviewer 1 Report
Thank you for the opportunity to review this manuscript. I commend the authors for conducting this important study that would be useful for public health policy. The large number of participants in this study is something worth commending.
I have some minor comments.
1. Under Data collection (i.e., 2.2), the authors mentioned data was gathered from 13 out of 14 regions of the Czech Republic. Please indicate the names of these 13 regions.
2. Under Measurement (that is 2.3.). please indicate the specific medical facilities used to measure weight and heights.
3. Please indicate the ethics number if Known under the ethcal consideration.
4. Please provide reference for the sentence " Our data from 2015 correspond with the COSI results and in both cases the difference is considerable"(Pages 289-290)
5.Given that the population for this study were children, was consent sought from parents? If yes, please indicate that. At the moment, it sounds like only consent was sought from participants (i.e. the children).
Reviewer 2 Report
Vazna et all present an interesting paper which highlights an important epidemic which is further worsened by the pandemic into light.
The manuscript is very well presented, opens with a good intro. The methodology and results are succinctly summarised.
There is a small point which if authors can address in a couple of lines (in discussion) would benefit the readers and perhaps may be helpful in further research on the topic viz:
What do the author think is the reason for the obesity trend to be most dramatic at a particular age (12 in this case) and starts descending afterwards? Is there any behavioural/psychosocial factor behind this?
Good luck to the authors for the publication.
Reviewer 3 Report
Review of the manuscript entitled: Influence of COVID-19-related Restrictions in the Prevalence of Overweight and Obese Czech Children. The manuscript submitted is appropriate to the subject matter and scientific rigor. The authors raised a very current issue at work, which is not only interesting from a scientific but also a practical point of view. Some remarks improving the quality of future research. and suggested changes and comments to the submitted manuscript in order to improve the quality of the planned research and future publications below:
1. Maybe in the chapter: Conclusions, authors should add digital data, not only describe the obtained results in words. Also the authors write in their conclusions about men and slowly developing hypogonadism as the article concerns children ????? explain it please or remove.
2. In the chapter Measurement please provide the name of the equipment and the manufacturer and references.
3. Would you please correct the bibliography in accordance with the journal's guidelines and standardize it. Put the full name of Journal or abbreviations, pages numbers and Issue. Please provide electronic access to first position in references [1] https://www.who.int/publications/i/item/origin-of-sars-cov-2
4. Would you please cheek it and correct references:
a) References [3] In reference [3] the results showed that:” During the COVID-19 confinement, PA (-91 ± 55 min/d, P < .001) and screen time (±2.6 h/d, P < .001) worsened, whereas the KIDMED score improved (0.5 ± 2.2 points, P < .02). The decrease of PA was higher in children with mother of non-Spanish origin (-1.8 ± 0.2 vs -1.5 ± 0.1 h/d, P < .04) or with non-university studies (-1.7 ± 0.1 vs -1.3 ± 0.1 h/d, P < .005) in comparison to their counterparts.”Why authors of this article provide other data. Please explain this ?
b) References [8] Authors write in this article:” Stress created by a bad financial situation in the family and a restricted social life could have led to craving comfort food that 60 usually contains excess sugar and/or fat [8]”. But in reference [8] we can read” „..briefly summarize contemporary research on food craving such as its neuronal underpinnings. They also highlight its relevance in obesity and binge eating disorder and suggest that research on and therapy of these disorders may benefit from providing an addiction framework. For instance, some approaches that effectively target drug cravings have also been shown to reduce food cravings. In a similar vein, Davis et al. (2014) investigated the effects of a methylphenidate challenge in individuals exhibiting addiction-like eating behavior. Individuals with “food addiction” reported more intense food craving than controls and were resistant to the food intake suppression that is typically induced by dopamine agonists. This supports that compulsive overeati”.
c) References [12] The aim of this study was to test the hypothesis that youths with obesity, when removed from structured school activities and confined to their homes during the coronavirus disease 2019 pandemic, will display unfavorable trends in lifestyle behaviors.” Here I can find information about „These factors occur predominantly during the summer, when it was proved that 66 children gain the most weight during the year”
d) References [13] In reference [13] is:” Results: From the fall of kindergarten to the spring of second grade, the prevalence of obesity increased from 8.9% to 11.5%, and the prevalence of overweight increased from 23.3% to 28.7%. All of the increase in prevalence occurred during the two summer vacations; no increase occurred during any of the three school years.” Would you please explain me Why you wrote „….the summer school holi- 68 days by 2-3%” I think it needs to be corrected?????
e) References[26] Would you please correct it: authors write:’ The average increase of weight was stated as 35%” Not weight gain but number of children who have been reported to gain weight: "..... Body weight increased in 35% of children / adolescents. A multiple regression analysis showed that the body weight increase was associated with increased consumption of breakfast, salty snacks, and total snacks and with decreased physical activity. The COV-EAT study revealed changes in ... "
f) Reference [27] Would you please check this reference in [27] we can read”…… An observational study found childhood obesity prevalence in the Philadelphia, Pennsylvania, region increased from 13.7% to 15.4% (2019-2020).2 The study included all patient visits and analyzed 2 time periods without a control period. Given obesity prevalence had been increasing prior to COVID-19……] correct it
g) Reference [35] Would you please check It it is less then 20% in reference[35]?
h) Reference [39] This article is about Infancy and we also can read „Recent data suggest that age at menarche may be static, but there is a debate as to whether the first signs of puberty are being seen much earlier in obese girls. Rapid early weight gain, obesity and early development may have implications for later health through the development of PCOS and overall association with cancer risk”. Would you please check this reference
i) Reference {42] concern research obese men not obese boys how wrote authors would you please check this reference or remove it or change for another
Also Authors write in their article (Wood et al 2016). I thin it is [19] position in authors reference Would you please correct the numbering in the references. See my comments below. Please provide the WHO references that you refer to in the article: '... of BMI was applied, in accordance with 135 WHO references for children. BMI categories were defined according to WHO"not items [11]from you reference
Remaining comments in the article in the attachment.
